# Simultaneous High-Speed Video Laryngoscopy and Acoustic Aerodynamic Recordings during Vocal Onset of Variable Sound Pressure Level: A Preliminary Study

**DOI:** 10.3390/bioengineering11040334

**Published:** 2024-03-29

**Authors:** Peak Woo

**Affiliations:** The Department of Otolaryngology, Head and Neck Surgery, Icahn School of Medicine at Mt. Sinai, 300 Central Park West 1-H, New York, NY 10024, USA; peakwoo@peakwoo.com

**Keywords:** high speed video laryngoscopy, voice onset, video kymomgraphy, phonatory function, evidence:N/A

## Abstract

Voicing: requires frequent starts and stops at various sound pressure levels (SPL) and frequencies. Prior investigations using rigid laryngoscopy with oral endoscopy have shown variations in the duration of the vibration delay between normal and abnormal subjects. However, these studies were not physiological because the larynx was viewed using rigid endoscopes. We adapted a method to perform to perform simultaneous high-speed naso-endoscopic video while simultaneously acquiring the sound pressure, fundamental frequency, airflow rate, and subglottic pressure. This study aimed to investigate voice onset patterns in normophonic males and females during the onset of variable SPL and correlate them with acoustic and aerodynamic data. Materials and Methods: Three healthy males and three healthy females were studied by simultaneous high-speed video laryngoscopy and recording with the production of the gesture [pa:pa:] at soft, medium, and loud voices. The fiber optic endoscope was threaded through a pneumotachograph mask for the simultaneous recording and analysis of acoustic and aerodynamic data. Results: The average increase in the sound pressure level (SPL) for the group was 15 dB, from 70 to 85 dB. The fundamental frequency increased by an average of 10 Hz. The flow was increased in two subjects, reduced in two subjects, and remained the same in two subjects as the SPL increased. There was a steady increase in the subglottic pressure from soft to loud phonation. Compared to soft to medium phonation, a significant increase in glottal resistance was observed with medium-to-loud phonation. Videokymogram analysis showed the onset of vibration for all voiced tokens without the need for full glottis closure. In loud phonation, there is a more rapid onset of a larger amplitude and prolonged closure of the glottal cycle; however, more cycles are required to achieve the intended SPL. There was a prolonged closed phase during loud phonation. Fast Fourier transform (FFT) analysis of the kymography waveform signal showed a more significant second- and third-harmonic energy above the fundamental frequency with loud phonation. There was an increase in the adjustments in the pharynx with the base of the tongue tilting, shortening of the vocal folds, and pharyngeal constriction. Conclusion: Voice onset occurs in all modalities, without the need for full glottal closure. There was a more significant increase in glottal resistance with loud phonation than that with soft or middle phonation. Vibration analysis of the voice onset showed that more time was required during loud phonation before the oscillation stabilized to a steady state. With increasing SPL, there were significant variations in vocal tract adjustments. The most apparent change was the increase in tongue tension with posterior displacement of the epiglottis. There was an increase in pre-phonation time during loud phonation. Patterns of muscle tension dysphonia with laryngeal squeezing, shortening of the vocal folds, and epiglottis tilting with increasing loudness are features of loud phonation. These observations show that flexible high-speed video laryngoscopy can reveal observations that cannot be observed with rigid video laryngoscopy. An objective analysis of the digital kymography signal can be conducted in selected cases.

## 1. Introduction

Direct observation of the vocal fold vibratory function is critical for understanding the physiology of phonation. Stroboscopy is the most practical instrument for recording and analyzing vocal vibratory function [1]. Stroboscopy uses the principle of stroboscopic light to analyze periodic vocal fold oscillations. A significant amount of information can be obtained by using this readily available tool [2,3]. However, stroboscopy cannot be used to evaluate rapid changes in vocal fold vibrations. One example is the onset and offset of vocal fold vibrations during phonation. The ability to use high-speed video laryngoscopy (HSV) and videokymography [4,5,6] allows the investigator to evaluate the vibratory patterns of individual vocal fold oscillations.

Voice initiation studies using high-speed cinematography were first reported in 1970 [7]. Since then, voice onset, offset, and diplophonia in normal and pathological states have been investigated using high-speed videos. In a report by Woo et al. using a rigid endoscope in 2017 [8], the phonation onset time showed significant differences between healthy and pathological states. Voice disorders such as spasmodic dysphonia, muscle tension dysphonia, and vocal fold scarring show a delay in the time between vocal fold adduction and steady-state vocal fold oscillation. Despite these observations, a refined understanding of vocal fold onset vibratory patterns with different vocal gestures in healthy “normal” individuals is yet to be achieved.

Previously, high-speed video laryngoscopy used a rigid 70° or 90° laryngoscope coupled with a high-speed camera. A 70° endoscope yields clear images; therefore, it is possible to analyze phonovibrograms. Phonovibrograms are waveforms that show individual cycles of vibratory patterns in high-speed video imaging and can be used to analyze vocal fold kinematics [9]. Rigid high-speed videos provide excellent image resolution but are not physiological. Running speech and natural voice onsets cannot be recorded using an oral laryngoscope.

The standard procedure is to acquire the video by asking the subject to phonate at a target pitch and loudness.

This approach does not allow observation of the tongue base, pharynx, and laryngeal height adjustments during vocal gesture production.

Fiberscope high speed imaging would be an ideal method for imaging of vocal fold vibration and such approaches have been published in prior studies [10]. Analysis of the fiberscope images were attempted [11].

Digital kymography tracings were obtained during natural speech; however, images recorded using this technique require enhancement and manipulation. Despite the use of image enhancement and noise reduction software, objective digital waveform analysis cannot be performed due to high noise.

Since 2015, more light-sensitive charge-couple devices capable of performing better high-speed videos have become available. The availability of a camera that can perform high-speed video laryngoscopy with a fiberoptic endoscope has several advantages. One advantage is that a fiberscope can be used through a face mask to perform simultaneous acoustic and aerodynamic sampling of various phonation gestures.

This study aims to use a new high-speed camera with an existing fiber-optic laryngoscope to investigate voice onset patterns while simultaneously recording acoustic and aerodynamic data. If adequate image resolution can be obtained simultaneously with the aerodynamic data, waveform analysis of the vocal vibratory patterns can be correlated with the acoustic and aerodynamic data. Simultaneous recording of visual and aerodynamic data using existing phonatory aerodynamic systems can be valuable for understanding normal and disordered voices.

This preliminary study reports the results of high-speed video recordings of the onset of vocal fold vibration in three healthy males and three healthy females. We varied the loudness from soft phonation to loud phonation. A high-speed video was recorded simultaneously with the acoustic and aerodynamic data sampled through an existing phonatory aerodynamic system. Using this approach, we aimed to analyze the subjective and objective observations of vocal vibratory function changes during vocal onset in the production of soft, medium, and loud phonations.

## 2. Materials and Methods

Three adult males and three adult females volunteered to participate in the study. All patients were deemed free of pathologies and lesions on prior examinations. One was an aspiring singer, whereas the others were English speakers without voice training. Instructions were provided using a phonatory aerodynamic system (PAS, Model 6600, Version 3.4.1; Pentax Medical, Montvale, NJ, USA). Fiberoptic laryngoscopy was performed using a high-speed camera. The recordings began once the subjects were able to produce the desired token with satisfactory repeatability. Figure 1 shows a photograph of the fiberscope with a high-speed camera attached to the PAS. To access the nose for fiber-optic laryngoscopy while simultaneously recording from the PAS system, a 3-mm hole was created in the facemask for insertion of the fiberscope and a seal was created at the entry site of the scope into the facemask using petroleum jelly.

### 2.1. Equipment Setup

#### 2.1.1. PAS

A voicing efficiency recording protocol was used in the PAS. The sensors allowed for the sampling of voice fundamental frequency, sound pressure level, mean airflow rate, and subglottic pressure. The token [pa:pa:] was tested at different SPL values. The source to microphone distance is set by the PAS system. Both systems were triggered, allowing high-speed laryngoscopy and PAS recording. Figure 2 shows a tracing of the recording from a female subject while voicing soft, middle, and loud voices.

#### 2.1.2. High-Speed Video Camera Setting

The cameras used for recording the high-speed videos were Chronos 1.4 and Chronos 2.1 high-speed cameras (Kron Technologies Inc., Burnnaby, BC, Canada; 32 Gigabyte monochrome system). The C-mount camera was attached to a 17.5-mm endoscope adaptor. The infrared filter of the high-speed camera was removed to maximize the light input to the sensor. The light source used to illuminate the larynx was a 300-watt xenon light source (Pentax Medical, Montvale, NJ, USA). High-speed images were sampled at 2000 frames per second with a resolution of 960 × 720 pixels. The shutter speed was 500 microseconds per frame, and the exposure time per frame was 494.44 microseconds. The shutter angle was set at 355°. Both analog and digital gains were +6 dB, and a 16-s sampling time was used.

#### 2.1.3. Flexible Laryngoscopy

A standard flexible laryngoscope was coupled with a high-speed camera for office laryngoscopy (Olympus P-4, Olympus Medical Lake Success, New York, NY, USA). The endoscope was a fiber optic laryngoscope and not a chip-tip scope.

### 2.2. Subject Recording

#### 2.2.1. Subject Instructions

Each participant produced three different tokens and each token was repeated twice. The three tokens were soft phonation at the modal pitch, the most comfortable voice at the medium loudness, and loud phonation at the modal pitch. Participants were instructed not to increase their pitch during a fiberscope examination. The target was to produce three phonation tokens at the most comfortable pitch with a 5-dB difference between each token of a soft, medium, and loud voice.

#### 2.2.2. Recording of Aerodynamic Data

An adult flow head transducer was used for all PAS flow recordings. The recordings were performed in a room with less than 55 dB ambient noise.

#### 2.2.3. Recording of HSV

A fiber optic laryngoscope was inserted into the nose after a local anesthesia spray. The scope was first inserted through a puncture in an aerophone mask and then threaded into the nose and pharynx. The scope tip was positioned at the epiglottis tip. The HSV camera buffer recorded the final 16-s of the video. The subjects produced tokens of soft, medium, and loud phonation of 1–2-s duration. Each participant produced each token twice, followed by the next token at the next loudness level. This was then followed by the loud token. The sequence used was soft-medium-loud phonation. During the production of the token, the fiberscope was advanced into the introitus of the larynx to visualize the false and true vocal folds.

Once the examiner verified token production, the trigger for video capture was activated. The video was then saved for subsequent analysis.

## 3. Analysis

### 3.1. Aerodynamic and Acoustic Analysis

There were variations within and between the tokens in the fundamental frequency, SPL, flow, and pressure. Therefore, averages of the parameters were taken from the two repetitions of each token.

### 3.2. Video Editing and Conversion

The original HSV recordings were in MP4 format with a resolution of 960 × 720 pixels. Changing the format to 320 × 240 AVI mode was necessary for video image analysis. A standard video editor was used to clip each video’s area of interest (typically 500–1000 frames). One thousand video frames captured 500-ms of the voice during the onset of the token, which comprises the onset of phonation from adduction to a steady state. The brightness, contrast, and gamma were adjusted to optimize the view of the vocal folds. The video was rotated such that the view of the vocal folds was vertical for subsequent image analysis. Video clips of the pre-phonation set, initiation of vocal fold vibration, and transition to steady-state phonation were clipped for digital image analysis.

A description of the video analysis for deriving the vibrogram waveform has been published [12]. The Kay Image Processing System (KIPS) program version 1.11 (Pentax Medical; Montvale, NJ, USA) was used for video image analysis. Video kymography of the mid-membranous vocal folds was performed using digital kymography (DKG). The digital kymogram was then converted into a vibrogram waveform by further adjusting the brightness and darkness to enhance the edges of the vocal folds. The edge-tracking software created a vibrogram based on the videokymogram. Once the edge tracking was complete, visual verification was performed to ensure its validity.

### 3.3. Vibrogram Analysis

The vibrogram analysis measured the time in frame numbers from the adduction to steady state. We also calculated the pre-phonation time from the onset of adduction to 1/3 of the width of the vocal fold to the beginning of oscillation and the number of frames needed to establish steady-state phonation.

Fast Fourier transform (FFT) analysis of the vibrogram waveform was performed for each videokymogram when possible. From the FFT plot, we recorded the power of the spectral peak at the fundamental and first three harmonics for each videokymogram. 

For tokens that could not be analyzed by digital waveform analysis owing to poor image quality, the DKG waveform was “read” by counting the approximate number of video frames between vocal fold adduction and vocal onset. Figure 3 shows a schematic of the steps from the HSV image acquisition to image analysis. Waveform analysis was used to calculate the power-spectrum values derived from the FFT plots. The spectral peak values from each vibrogram waveform can be used to compare different tokens. Thus, a high spectral peak at the fundamental frequency suggests a greater amplitude of vocal fold oscillation than one that has low amplitude. A spectral peak that is flat and poorly defined suggests frequency splaying when compared with a sharp well defined peak [13].

## 4. Results

Simultaneous voice-onset recordings of high-speed video, aerodynamic, and acoustic data were recorded in six subjects voicing at different SPL (in dB) levels. Using the recording fiberscope device through the PAS mask, it was possible to obtain a high-speed video of the flexible laryngoscopic examination of the larynx while simultaneously sampling acoustic and aerodynamic data.

### 4.1. Aerodynamic Changes during the Production of Different SPL

Figure 2 shows a PAS display of the tokens of phonation from soft to loud voice onsets for one subject. The data for each token of each participant are listed in Table 1. The table includes the subject, sex, age, average sound pressure level (dB), frequency (Hz), airflow (cc/s), and subglottic pressure (cm H_2_O) for each token. The data tabulated are an average of the two tokens produced. The calculated values derived from these four measures were also tabulated. These include the resistance ((pressure/flow) × 1000), decibel change between tokens, resistance change between tokens, and resistance change per dB between tokens. The average increase in SPL from soft to loud phonation was 15 dB. The median phonation fundamental frequency for the male group was 119+/−13 Hz and 189+/−24 Hz for the females. As expected, there is a steady increase in the subglottic pressure with increased SPL (9.7 cm H_2_O for loud, 5.7 cm H_2_O for medium, and 3.9 cm H_2_O for soft). The resistance was the highest for loud phonation. The flow varied between tokens. Soft phonation had the lowest flow rate, medium phonation had the highest mean airflow rate, and the loud phonation airflow rate value was higher than that of soft phonation but less than that of medium phonation. Table 2 summarizes the mean and standard deviation values for the frequency, flow, and pressure of tokens produced by females and males. The resistance increase is significantly higher per decibel for the middle-loud token than that for the soft-middle token. Thus, for the production of loud phonation, a higher glottal resistance is needed compared to medium phonation. For soft-to-modal phonation, a lower resistance per decibel increase was observed. Higher resistance to middle-loud phonation was observed in both females and males (Table 1).

Aerodynamic analysis of vocal onset tokens showed differences between the tokens.

When moving from soft to middle phonation, no significant change in the glottal resistance was required. There was a slight increase in the mean resistance per decibel from the soft to middle voice. When the subject produces loud phonation, there is a higher glottal resistance, with a mean resistance per decibel significantly higher than that of soft-middle phonation. Table 3 lists the different subjects and resistance change per dB change in the produced tokens from soft to middle to loud. Figure 4 shows a graphical representation of the change in resistance per dB for the six subjects for soft-to-middle and middle-to-loud voices. The necessary change in the resistance per decibel was greater for middle-loud phonation than that for soft-middle phonation. These changes were significant (*p*-value < 0.01, paired *t*-test).

### 4.2. Descriptive Findings of High-Speed Video of Vocal Onset

Figure 5, Figure 6 and Figure 7 show the video montages of each video for female Subject 4 for the three tokens. Figure 5 shows the montage of the voice onset for soft phonation (212 Hz, 76 dB). Figure 6 shows a montage for medium phonation (215 Hz, 85 dB). Figure 7 shows a montage for loud phonation (217 Hz, 90 dB). In each instance, the vocal folds were approximated toward the midline before the onset of the vibration. For each figure, the down arrow indicates a just noticeable vibration and the second arrow the beginning of steady state oscillation. Vibrations occur before full vocal fold closure. After steady oscillation is achieved, there is a typical glottis configuration for the intended fundamental frequency and SPL. With increasing SPL, some adjustments were made to the supraglottic posture to achieve the intended SPL. Typically, three to four glottal cycles are necessary to achieve a soft phonation steady state. A similar finding was observed in medium phonation with a more pressed inward movement of the false fold. The vocal folds seem to shorten as time progressed (Figure 6). These changes occurred during the first 50 ms of phonation. With loud phonation, more frames showed glottal and supraglottic adjustments before reaching a steady state. (Figure 7). In this example, almost twice the onset time from noticeable phonation to steady state was observed compared to that with soft phonation.

Vocal folds started vibrating without requiring vocal fold closure in four subjects, regardless of the token. Two subjects (Female 6 and Male 1) showed muscle tension patterns with loud phonations. These subjects initiated a voice with a base of tongue tension by the tongue going backward, thus obliterating the view of the larynx during the initial onset of phonation. Adjustments in tongue tension allow the larynx to return to view. Thus, laryngeal vocal fold oscillations could not be tracked between gestures, because with loud phonation, the view of the vocal folds was obliterated. When there was epiglottis retroflection, there was a whiteout of the DKG and a reappearance of the DKG signal, as shown in Figure 8. The voice onset time was prolonged. Figure 9 shows the DKG tracing of medium and loud phonation (Female 3, 207 Hz, 92 dB) onsets in subjects with complete vocal fold adduction. It can be seen that the vocal folds come to complete closure before the onset of phonation.

In this example, there was a long prephonation time delay, but the number of glottal cycles required to establish a periodic vibration was short. In this example, only four cycles of vocal fold oscillations were required before steady state was reached. While soft and medium phonation show a gentle approximation to the midline with a predictable duration of several cycles before a steady state of vocal fold oscillation is achieved, such predictability cannot be generalized to loud phonation. Variability is observed in the production of loud gestures. This is more consistent with medium phonation. For example, Figure 10 shows the DKG results for three male subjects during medium phonation. All patients exhibited anterior-to-posterior squeezing with short vocal folds. In all three cases, the vibration requires 3–5 glottal cycles before reaching a steady state. Furthermore, there was no evidence of complete closure of the glottis before the onset of vocal fold vibration. The need for supraglottic adjustment before the onset of phonation was more obvious in males than females.

### 4.3. DKG Tracing and Analysis of the Vibrogram Waveform

Although we obtained DKG tracing for the subjects using flexible laryngoscopy, objective edge-tracking software could derive adequate waveforms for analysis in the females but not males. Image quality is a primary issue. The images were difficult to capture in males due to the higher supraglottic squeezing seen in the male subjects, or it may be due to the male larynx being deeper and farther away from the tip of the laryngoscope. When the images were highly grainy, edge-tracking was insufficient to derive a waveform for objective analysis. For the male subjects, the DKG plots were evaluated visually. We successfully performed edge tracking and waveform analysis in females.

Figure 10 shows the edge-tracking software for Subject 4 with soft, medium, and loud phonations. The edge-tracking software performed credible edge tracking and plotted the edge as a vibrogram waveform. The vibrogram waveform can be transformed into FFT plot to analyze the vocal fold oscillation signal properties at different SPL values, as shown in Figure 11. With soft phonation, the sinusoidal pattern of vocal fold oscillation results in peak signal power at the fundamental frequency. No significant energy was observed at higher harmonics. With loud phonation, a stronger energy peak was observed at the fundamental frequency, as well as the first and second harmonics. This harmonic energy increase is consistent with the observation of short open and longer closed phases of vocal fold oscillation. These higher spectral energy peaks indicate a higher amplitude of vocal fold oscillations and faster opening and closing of the vocal folds with loud phonation [13]. The higher energy peaks and energy at higher harmonics were consistent with the visula observation of higher velocity change and acceleration of the vocal folds during loud phonation. There was a higher amplitude of vocal fold oscillations with loud phonation than with soft or medium phonation. Larger spectral peaks with loud phonation were a consistent finding, as shown in Figure 12 and Figure 13 for the two female subjects. These figures show an increase in the F0 energy with an increase in SPL. Increases in F1 and F2 energies with increasing SPL were also observed.

## 5. Discussion

This is a preliminary study on a technique to obtain HSV simultaneously with physiological data for the study of vocal fold function during voice onset.

The observation of the larynx during phonation and speech has led to significant advances. Advances include narrowband imaging [14], high-speed video laryngoscopy [15], chip-tip videolaryngoscopy [16], videokymography [17], and computerized videostroboscopy [3]. Functional assessment of vocal output includes measuring aerodynamic data [18], voice range profile [19], phonetogram [20], and self-assessment of voice handicap [21]. These are tools for assessing vocal pathology and response to treatment. The PAS system is one such tool that collects data on sound pressure, frequency, subglottic pressure, and airflow. The PAS can be used to document vocal efficiency and resistance in various vocal tasks [22]. Researchers and clinicians use the PAS to document changes in flow and resistance to surgery and medical treatments. Another increasingly useful tool for imaging vocal fold vibrations is high-speed laryngoscopy. High-speed video laryngoscopy (HSV) can examine each glottal cycle. It can be orally or nasally administered [23]. HSV is valid for studying non-periodic vibrations such as voice breaks, diplophonia, or aperiodic oscillation sources in the vocal tract. Unlike stroboscopy, the quantitative assessment of vibratory patterns is not dependent on a quasi-periodic oscillatory pattern [24]. HSV is ideal for objectively studying the onset and vibratory patterns of different vocal fold oscillations. Rigid endoscopy for voice onset has been reported in research [25] and in normophonic participants [9]. This investigation simultaneously acquired HSV videos with aerodynamic and acoustic data.

Voice onset requires the approximation of folds towards the midline; however, it does not require a complete approximation. The pre-phonatory set of the larynx and its adjustment between approximation and the steady state of oscillation for the intended gesture lends itself to observation by high-speed video endoscopy [26]. In a study of normophonic participants and pathological vocal folds using rigid endoscopy [8], observations revealed that vocal onset was markedly delayed in several pathological disorders. Two types of vocal onset delay were noted. Pre-phonation delay is when the vocal folds are nearly approximated, and the time to the first observed oscillation is prolonged. A steady-state delay occurs when there is prolonged vocal folds oscillation until a steady state is observed. In these clinical studies, the SPL or fundamental frequency was not controlled.

Vocal onset in response to various SPL targets is a particularly relevant clinical problem. Loud phonation is more likely to affect vocal fold function. It has been shown to increase shear and impact stress, thereby promoting vocal fold injury. An observation in routine high-speed laryngoscopy, loud phonation is often accompanied by extra-laryngeal gestures such as pharyngeal squeezing, base of tongue tension, and anterior-to-posterior vocal fold shortening [27]. These observations of excess muscle tension are not readily appreciated by rigid endoscopy but can be easily documented by flexible HSV.

Flexible high-speed laryngoscopy presents technical challenges. It is difficult to obtain good images because of the limited light-carrying capacity of the small endoscope. Unlike rigid endoscopes, the light transmitted through the light bundle using a 300-watt bulb is still quite attenuated. In the past, post-acquisition image enhancement and image resolution were necessary [11]. Such technical limitations in sensor technology have rendered fiber optic HSV observations impractical. With recent technical improvements in image sensor resolution and lower costs of commercial high-speed cameras, practical observations using handheld high-speed imaging devices can now be performed. The HSV camera was coupled to a standard fiber optic laryngoscope. Challenges in light sensitivity and acquisition time can be overcome by increasing the light shutter time and removing the infrared filter while keeping the focal length of the camera small. These adjustments allow for real-time imaging, acquisition, and playback without requiring post-recording image enhancement.

The challenges of simultaneously recording high-speed images with an aerodynamic mask in place can be overcome by adjusting the aerophone mask. A skilled endoscopist is necessary to obtain the tip of the scope close enough to the vocal folds for post-hoc image analysis. Without a close-up view of the vocal folds, the images were too gray and lacked detail. The video’s pixel counts encompassing the vocal folds were too small for edge detection using the KIPS software. Although imaging of vocal fold vibratory behavior can be visually analyzed, and a DKG waveform can be generated, objective waveform analysis is not possible. Thus, the examiner must advance the scope close enough to the vocal folds so that the vocal folds nearly fill the circle of the endoscope during imaging. For imaging, it is reasonable to start by placing the fiber-optic laryngoscope above the epiglottis to evaluate supraglottic adjustments and then advance the scope to acquire images of the vocal folds. The scope was advanced approximately 5 mm away from the vocal folds. 

A 32-gigabit buffer allowed for 16 s of sampling time. This sampling time is generally sufficient to acquire the area of interest for post-recording analysis.

An observation of note is the amount of extralaryngeal functional adjustments in the vocal tract during token [pa:pa:pa:] production. The [pa:pa:pa:] gesture required significantly more supraglottic motion than that when the subject was asked to have an easy voice onset, such as when using the token [ha:ha:ha:] To estimate the subglottic pressure, the token [pa:pa:pa:] is necessary to obtain bilabial closure before voicing. However, this gesture is accompanied by tongue tension, posterior tilting of the epiglottis, and pharyngeal squeezing. In two of the six subjects, extra-laryngeal tension made visualization of the laryngeal introitus impossible.

We tried to obtain voice onset time for the various gestures. This would have required tabulation of the time from glottis adduction to full steady state oscillation. This was not practical. The reasons are two-fold. First, there were multiple examples during onset when the vocal folds could not be seen due to muscle tension, thus making the image of the edge of folds impossible. Second, with only six subjects sampled at three different loudness levels, the descriptive findings are the best that can be tabulated. Despite the limited ability to record satisfactory videos from all subjects for image analysis, satisfactory images can be obtained with sufficient detail to analyze voice onset using DKG. DKG tracing provides good documentation of the onset of vocal fold vibrations for a specific token. The DKG can be read to determine the duration of the time needed for the prephonation onset delay and the time required to reach steady-state oscillation. In this series, loud phonation resulted in the most significant increase in resistance. The increase in resistance per decibel was much higher than the increase in resistance during the progression from soft to medium phonation. With a loud token, there is prolonged delay from vocal fold closure to steady state. Although the vocal folds are brought to an approximation early, several cycles are required to adjust to steady-state phonation. The prolonged adjustment time is shown in Figure 5, Figure 6 and Figure 7. Higher collision forces are observed in loud phonation, which is supported by the observation on FFT spectra of greater spectral energy at the higher harmonics. There is higher amplitude of each spectral peak with louder phonation. Higher spectral peaks in the first and second harmonic with loud phonation indicate more rapid change in the configuration of the vocal folds, implying greater acceleration and deceleration of the vocal folds with loud phonation. Loud phonation is associated with an increase in amplitude and longer closed time of the vocal folds.

The simultaneous recording of vocal gestures with acoustic-aerodynamic data can be helpful as a research tool for investigating vocal physiology. Recording pharyngeal adjustments during the onset of phonation is challenging with a rigid endoscope. We were unable to track pharyngeal adjustments objectively using the existing imaging software. Using existing software, one can perform edge tracking to evaluate vocal fold oscillations during different vocal gestures during voice onset.

The ability to record pressure, airflow, and acoustic parameters simultaneously may be helpful for further investigations of phonation physiology. In the future, by multiple testing in normal and pathological situations using this approach, we may gain an understanding of the optimal voice onset and voice exposure. This implies that the increase in resistance with increasing SPL is not linear. With more subjects, the aerodynamic data can be correlated with the spectral correlates of vocal fold vibrations with different vocal gestures. Understanding the acceleration and deceleration of the vocal folds with acoustic and aerodynamic correlates can help determine the most efficient way to produce sounds without vocal damage.

## 6. Conclusions

This is a preliminary study on a technique to allow simultaneous acquisition of nasoendoscopic HSV, acoustic, and aerodynamic data.

In selected cases, a high-speed video with sufficient fidelity can be acquired to objectively analyze the vocal vibratory pattern. Using this approach to evaluate soft, medium, and loud vocal onsets, we showed that increasing glottal resistance with the onset of different target SPL was not linear. Furthermore, the onset of voicing is accompanied by pharyngeal adjustment. These compensatory gestures occur in the extra-laryngeal vocal tract and at the vocal fold level. These observations reflected the reality of vocal production in non-encumbered subjects. Future evaluations of vocal physiology should include visual, acoustic, and aerodynamic data to understand physiological function in both healthy and pathological states.

## Figures and Tables

**Figure 1 bioengineering-11-00334-f001:**
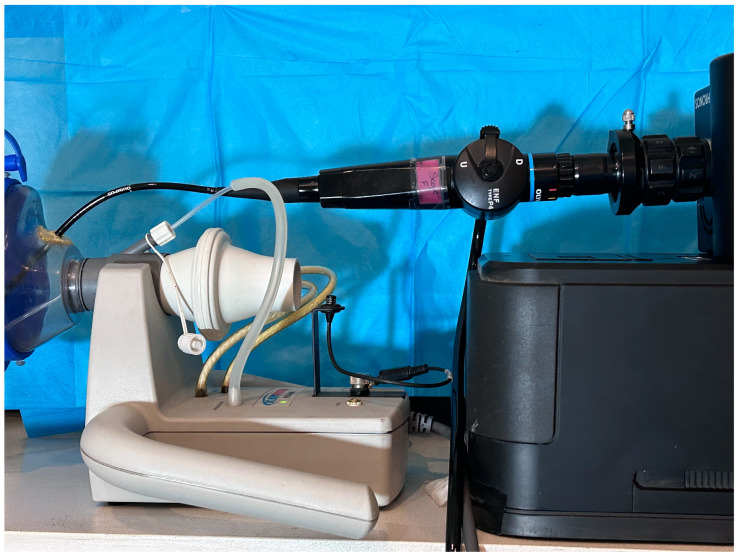
A photograph of the fiberscope with the high-speed camera attached to the PAS system.

**Figure 2 bioengineering-11-00334-f002:**
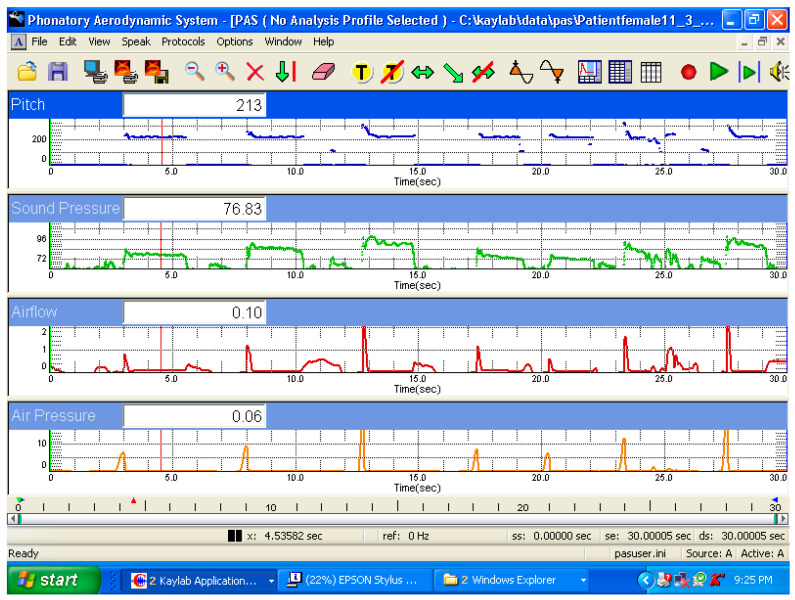
Phonatory aerodynamic system (PAS) tracing of the recording from a female subject during the voicing of soft, middle, and loud voice production. Time is on the X axis. The top tracing is the frequency, the second from top is the sound pressure level (SPL), the third tracing from the top is the airflow in cc’s/s, the bottom tracing is the intra oral pressure tracing in cm H20.

**Figure 3 bioengineering-11-00334-f003:**
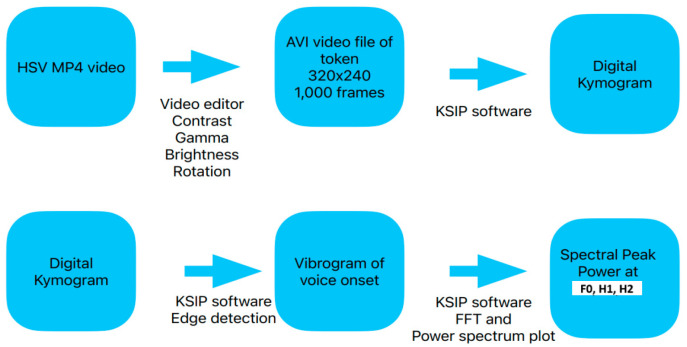
Schematic of the steps done from image acquisition of the HSV to image analysis.

**Figure 4 bioengineering-11-00334-f004:**
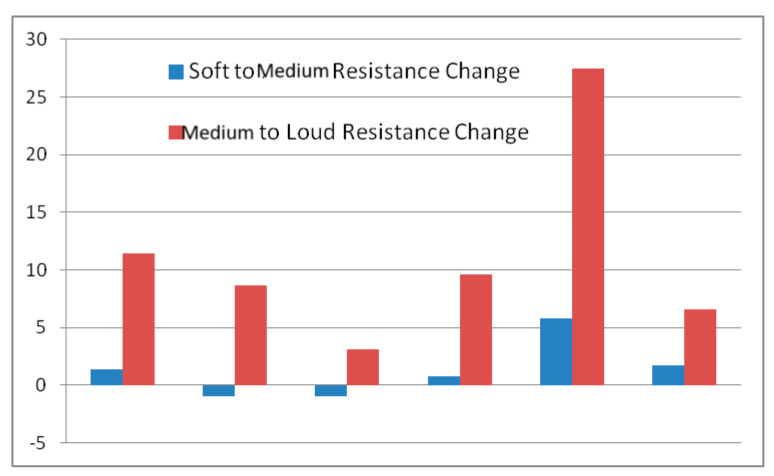
Plot of the changes in resistance per dB for the six subjects for soft to middle voice and middle to loud voice.

**Figure 5 bioengineering-11-00334-f005:**
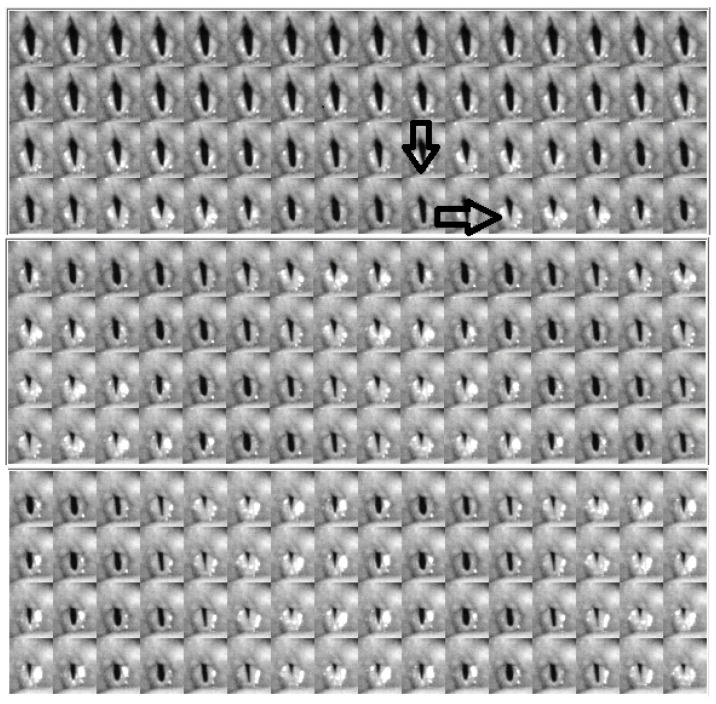
Video montage of voice onset for soft phonation (212 Hz, 76 dB), female, subject 4. Right arrow indicates just noticeable onset of oscillation. Down arrow indicate beginning of steady state oscillation.

**Figure 6 bioengineering-11-00334-f006:**
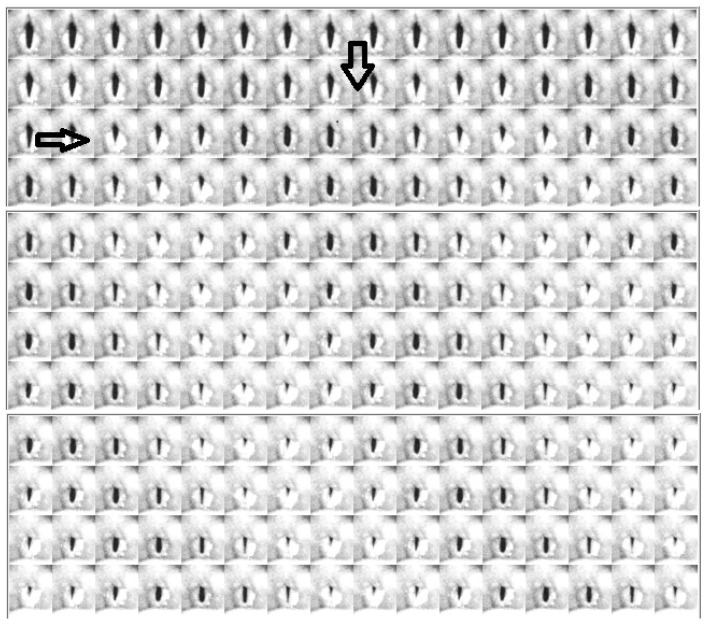
Video montage of voice onset for medium phonation (215 Hz, 85 dB), female, subject 4. Right arrow indicates just noticeable onset of oscillation. Down arrow indicates beginning of steady state oscillation.

**Figure 7 bioengineering-11-00334-f007:**
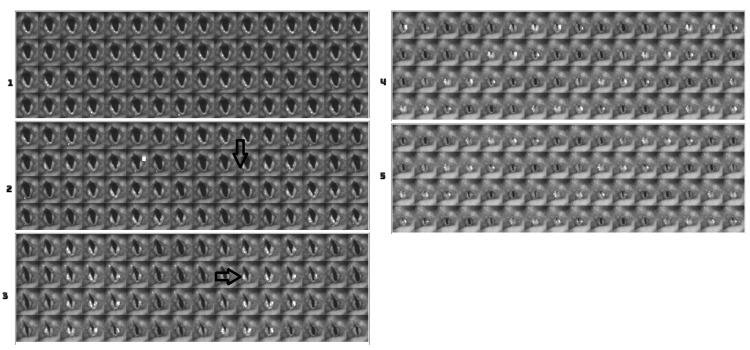
Video montage of voice onset for loud phonation (217 Hz, 90 dB), female, subject 4. Right arrow indicates just noticeable onset of oscillation. Down arrow indicates beginning of steady state oscillation.

**Figure 8 bioengineering-11-00334-f008:**
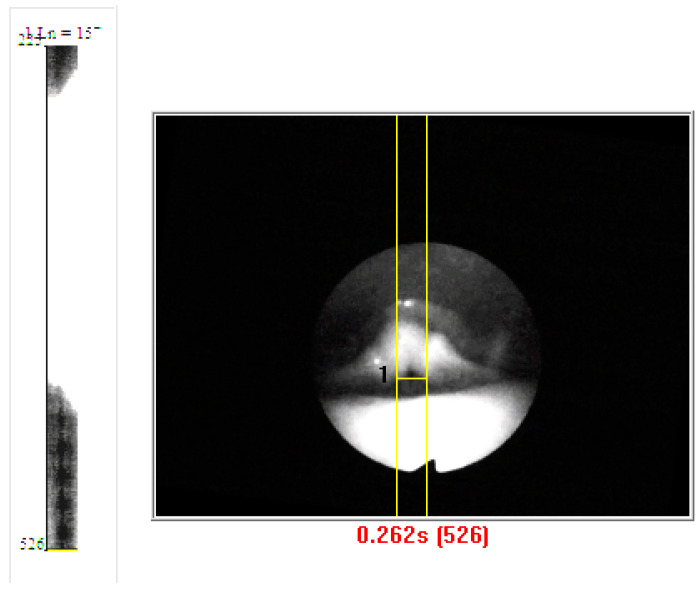
DKG Showing base of tongue tension that obliterated the view of the larynx during the initial onset of phonation.

**Figure 9 bioengineering-11-00334-f009:**
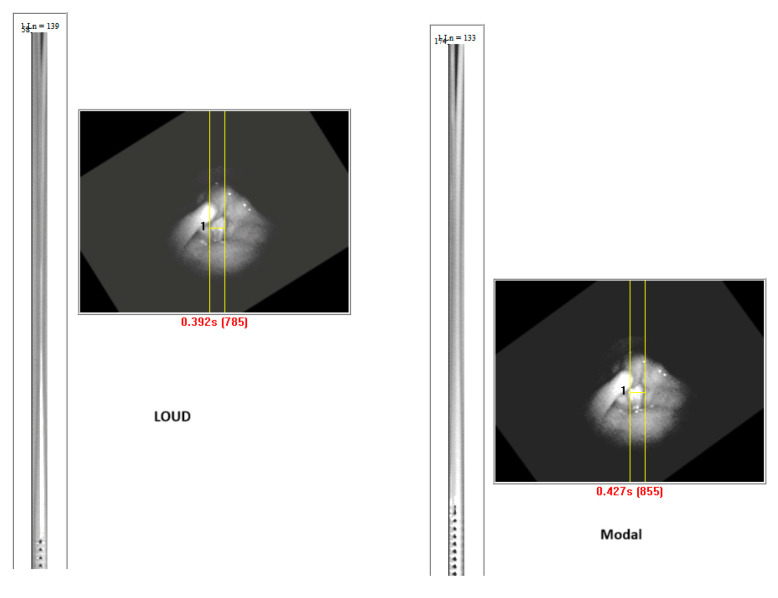
DKG tracing of middle and loud phonation (female 3, 207 Hz, 92 dB) vocal onset with full vocal fold adduction. Note the long closing time before phonation onset.

**Figure 10 bioengineering-11-00334-f010:**
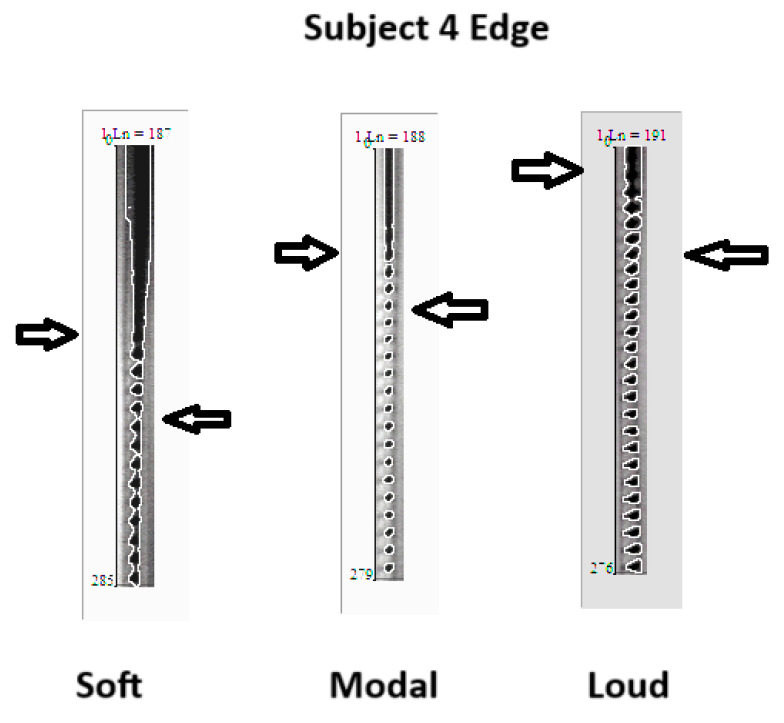
DKG tracing of and edge tracking for subject 4 with soft, middle, and loud phonation. Right arrow is the just noticeable vibration. Left arrow is the beginning of steady state of vocal fold oscillation.

**Figure 11 bioengineering-11-00334-f011:**
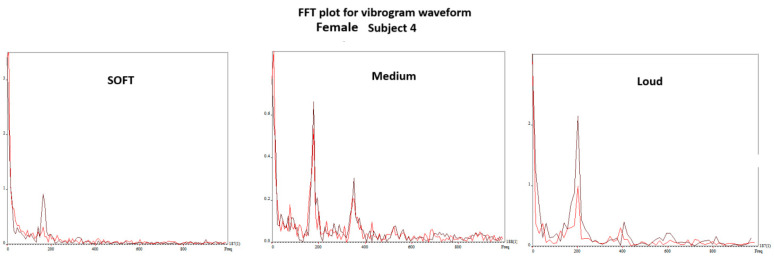
Vibrogram FFT plot of vocal fold oscillation onset at soft, middle, and loud phonation (subject 4).

**Figure 12 bioengineering-11-00334-f012:**
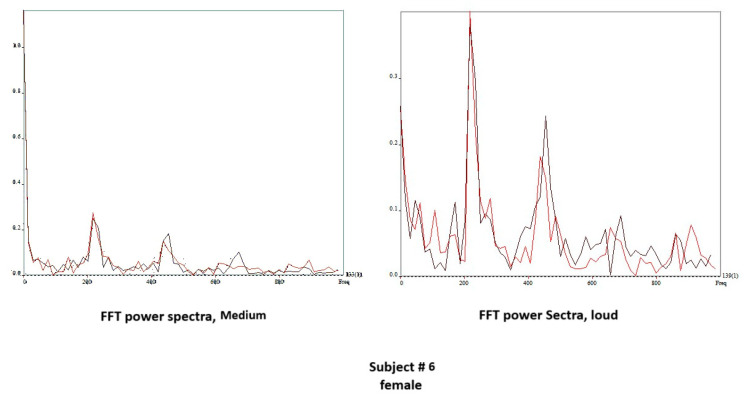
Vibrogram FFT plot of middle versus loud phonation (Subject 3).

**Figure 13 bioengineering-11-00334-f013:**
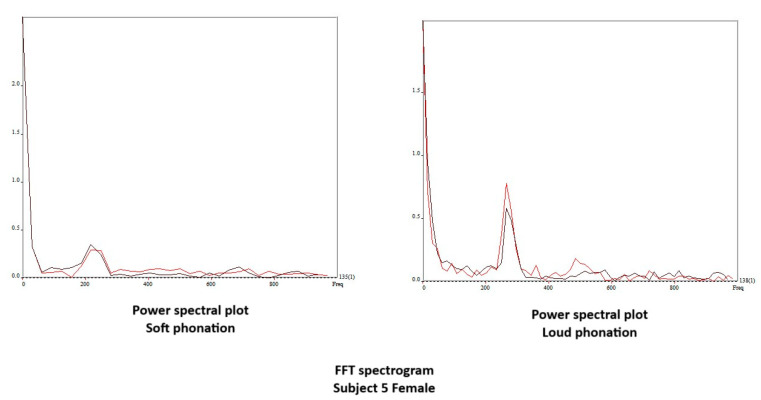
Vibrogram FFT plot of middle versus loud phonation (Subject 5).

**Table 1 bioengineering-11-00334-t001:** Tabulation of aerodynamic and acoustic data for the six subjects.

Subject	Sex	Age	Token	SPL dB	Frequency Hz	Flow cc/sec	Pressure cm	Resistance Pressure/Flow	dB Change	Resistance Change	Resistance Change per dB
1	male	33	soft	74	110	210	3	14.28571			
1	male	33	modal	85	109	170	5	29.41176	11	15.12605	1.375095
1	male	33	loud	88	110	110	7	63.63636	3	34.2246	11.4082
2	male	29	soft	70	102	79	4	50.63291			
2	male	29	modal	83	121	132	5	37.87879	13	−12.7541	−0.98109
2	male	29	loud	89	139	89	8	89.88764	6	52.00885	8.668142
3	male	69	soft	85	125	100	2	20			
3	male	69	modal	91	123	220	3	13.63636	6	−6.36364	−1.06061
3	male	69	loud	100	136	170	7	41.17647	9	27.54011	3.060012
4	Female	64	soft	74	161	100	4	40			
4	Female	64	modal	84	157	110	5.3	48.18182	10	8.181818	0.818182
4	Female	64	loud	87	186	130	10	76.92308	3	28.74126	9.58042
5	Female	24	soft	76	212	100	6	60			
5	Female	24	modal	85	215	80	9	112.5	9	52.5	5.833333
5	Female	24	loud	90	217	60	15	250	5	137.5	27.5
6	Female	30	soft	75	168	130	4.6	35.38462			
6	Female	30	modal	86	182	130	7	53.84615	11	18.46154	1.678322
6	Female	30	loud	92	207	120	11.2	93.33333	6	39.48718	6.581197

**Table 2 bioengineering-11-00334-t002:** Mean and Standard deviation values for frequency, flow and pressure for the tokens produced for females and males.

		Frequency Hz	Flow cc/sec	Pressure cm H_2_O
Mean	Male	119	142	4
SD		12	52	2
Mean	Female	189	106	8
SD		24	24	3

**Table 3 bioengineering-11-00334-t003:** Resistance change per dB for the six subjects for soft to modal phonation and modal to loud phonation.

	Soft-Modal Resistance/dB	Modal-Loud Resistance/dB
Mean	1.27	11.15
STD	2.49	8.5
Paired *t*-test	*p*-value < 0.01	*p*-value > 0.01

## Data Availability

Available by contacting the author peakwoo@peakwoo.com.

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
