# Peer review of "Simultaneous High-Speed Video Laryngoscopy and Acoustic Aerodynamic Recordings during Vocal Onset of Variable Sound Pressure Level: A Preliminary Study"

_bioengineering, 2024, doi:10.3390/bioengineering11040334_

Round 1

Reviewer 1 Report

Comments and Suggestions for Authors

The manuscript describes a multi-methodic study of phonation onset. Vocal fold vibration is studied with high-speed camera using a nasofiberoscope. Simultaneous recordings of acoustic signal, and airflow and airpressure are made. The topic is very relevant, as the onset of phonation may offer valuable information of the state of the vocal folds and also of the phonation type used. The recording system is described quite well, and the description and the preliminary findings have clinical value. However, the manuscript requires a major revision before it can be accepted for publication. In its present form the text appears to be written in a hurry, there are various typos and awkward sentences. It is also somewhat confusing for the reader that the topic is onset of phonation but the acoustic and aerodynamic results are presented for the whole syllables, as it appears, not related to onsets as such, and results of the onsets are shown in a more descriptive way, e.g. without a table summarizing the onset durations. From the figures it is not easy to see where exactly the onset ends and steady phonation starts. No correlation is made with the image results and acoustic and aerodynamic results.  The manuscript is more like a description of a method rather than an experimental study of a specific topic. It is also entitled to be a preliminary study so certain deviations from a full experimental investigation are tolerable. However, in my opinion the manuscript requires improvement. Below is a list of more detailed remarks which I hope to be helpful for the author to improve the text.

More specific remarks

Firstly, the text requires proofreading by a native speaker of English.

Abstract

Line 15 :“loudness and frequencies” needs revision. Loudness is a perceptual term, frequency is a physical one. I recommend that you stick to physical terms since you are measuring here physical variables. PAS seems to use loose terminology (like Praat software) when labelling ‘pitch’ when a more accurate label would be fundamental frequency (f0) (since pitch is not measured in Hz but e.g in mel).

Line 16: “by per oral endoscopy…” I am not a native speaker of English but this expression sounds awkward to me. Perhaps it could be: endolaryngoscopic investigations using a rigid endoscope inserted through the mouth…

Line 20: Flow is here and elsewhere written with capital letter. Please change to ‘flow’ or better: airflow.

Lines 18-20: “We have adapted…” the sentence seems to have some repetition. Please improve.

Line 18: Should it be adobted instead of “adapted”?

Line 24: “soft, medium and loud modal voice.” What do you mean by modal voice? Voice in modal register? Would it be clearer to say: soft, medium and loud speaking voice?

Line 24: The marking //PAA// etc. seems strange to me. I suggest [pa:pa:pa:] or repetition of the syllable /pa:/ three times.

Lines 24-25 and elsewhere: I recommend fiber-optic.

Lines 24-26: “The fiber optic…” The sentence sounds a bit strange. Would it be clearer with “to allow simultaneous…”?

Lines 26-27: “The average increased SPL for the group was 15 dB from 70 dB to 85 dB.” I suggest something like: The average increase in SPL … Or: SPL increased in average…

Line 27: Instead of “Flow was increased…” I suggest: Flow increased.

Line 28: I suggest to use past tense when reporting about the results, e.g.: There was a steady…

Line 30: “modal to loud”. ? Should it be medium (loudness)?

Lines 30-32: “High- speed video analysis using videokymogram showed the onset of vibration in all loudness tokens without needing full glottis closure.” This sentence is not clear. In general, please, define more clearly how exactly the onset of vibration was measured.

Lines 36-37: Compensatory effect? Compensation for what?

Line 40: “Vibration analysis of voice onset shows that more time is necessary before oscillation stabilizes to a steady state”. More than in what?

Line 43: Is this a typo: “pr-phonation time”?

Introduction

Line 74: “These phonovibrograms can analyze…” I suggest: It is possible to analyze using….

Lines 77-79: “The standard token is by asking…” I think that this sentence requires reformulation. Maybe: token is obtained by asking… And here: at a target pitch and loudness (perceptual terms as you are dealing with humans who control their voicing by perception).

Lines 94-95: “Suppose…” ? Is there something missing in this sentence?

Line 96: “correlate those”? Should it be: it?

Line 106: “mobile”? Should it be medium?

Material and Methods

Please tell how the acoustic signal was recorded and what the mouth to microphone distance has been.

Figure 2 gives an overall view on the samples recorded, but how about showing and measuring such values for the onsets? Furthermore, the text tells that each subject produced three times the syllable [pa] for each of the three loudness levels. This does not seem to be the case that is shown in Figure 2. It might be nicer to see three syllables per each loudness level, if possible. Was there in all cases so much variation in the airpressure from one [p] to the next as it seems to be in Fig. 2?

Line 128: “soft, middle, and loud”, lines 163-164:”soft, modal and loud…” Please, check the terms. Sometimes the medium loudness seems to be labelled with medium, sometimes with modal, sometimes with middle.

Line 172 and elsewhere: “data is”, please notice that data is plural, so data were…

Line 173: typo: Db – should be dB.

Line 176: “native HSV recording…? Would it be better to say original?

Lines 206-207, the sentence contains repetition, please revise.

Figure 3 and elsewhere: The notations require revision. Capital letters F1 and F2 refer to formant frequencies, not to harmonics. The recommended marking would be either to use H1, H2 (H meaning harmonic) or: 2fo, 3fo etc. according to Titze et al. 2015 (notice that here fo is written with o and not with zero).

Line 214, typo: highs speed…

Line 217: The sentence is imperfect.

Lines 224-225 and elsewhere: Please, notice that loudness is not measured in dB, since loudness is a perceptual term (it is measured in sones and phones). Please use the physical term sound pressure level. (You may say e.g. that when increasing loudness the subjects raised SPL…).

Line 228: Please write more exactly: 9.7 cm H2O etc.

Line 231: Should it be smaller (instead of larger) here?

Lines 234-236: The sentence requires improvement.

About the results in Table I: The air pressure values seem to be small to represent ‘loud’, since 7 cmH2O is typically observed in ordinary speaking.

Line 248: Should be producing…

Lines 250-251:  “much more significant…” What does this mean? I mean, from what this conclusion comes?

Line 261 – and elsewhere, there is typo:  212(Hz.

Line 269: “Some vocal fold shortens…? Please, modify the sentence.

Figure 4. I wonder how a change in resistance can be negative. Do I misunderstand here something?

Figures 5-7: In my opinion, readers need some guidance in here. Could you perhaps add some arrows showing where you think that the onset is completed.

The DKG tracings may not be familiar for all readers. Maybe you could add one for just showing how to read the tracing in general – and then, again here in Fig. 10, could you add an arrow to tell where the onset is completed. Or are these tracings only from the beinning to the end of onset?

Figure 11: Please add information what red and black lines represent. Furthermore, please explain more clearly what your aim is in taking FFT from the images.

Table II is not necessary, I think. You could just tell the results within the text.

Line 320: “Figure 10 is the edge tracking software…” The sentence sounds strange, please modify.

Line 330:”…there are greater spectral energy peaks…” What does this mean? Does “greater” refer to stronger or higher in frequency or maybe that there are more harmonics visible? Furthermore, the next sentence starts: “This higher spectral energy peak…” Just one? Or more of them since the preceding sentence told about “peaks”?

Lines 334-335:”More extraordinary spectral peaks”…? What does this mean? How are they extraordinary?

Lines 366-367: “The simultaneous recording of aerodynamic data simultaneously as high-speed video-laryngoscopy has not been done before.” This is not correct. Mehta et al. have published such data already in 2015. (Additionally, please revise the sentence, e.g. simultaneous recording of aerodynamic data with …)

Line 387: “shear stress” ? Do you really mean shear stress? Much more common is to relate vocal fold loading with impact stress.

Line 410: “A sampling time of 32 gigabits”? This sounds a strange formulation as gigabit is a measure of data amount, not time as such.

Line 441: Typo: However, the …”

Discussion could benefit from adding more references to previous studies about vocal onsets,  vocal fold loading and changes in resistance as a function of changes in SPL

References:

Titze, I. R., Baken, R. J., Bozeman, K. W., Granqvist, S., Henrich, N., Herbst, C. T., ... & Wolfe, J. (2015). Toward a consensus on symbolic notation of harmonics, resonances, and formants in vocalization. The Journal of the Acoustical Society of America, 137(5), 3005-3007.

Mehta, D. D., Deliyski, D. D., Zeitels, S. M., Zañartu, M., & Hillman, R. E. (2015). Integration of transnasal fiberoptic high-speed videoendoscopy with time-synchronized recordings of vocal function. Normal & Abnormal Vocal Folds Kinematics: High Speed Digital Phonoscopy (HSDP), Optical Coherence Tomography (OCT) & Narrow Band Imaging (NBI®), 1, 105-114.

Comments on the Quality of English Language

The manuscript requires checking by a native speaker of English.

Author Response

Please find the response in the attachment

Reviewer 2 Report

Comments and Suggestions for Authors

Review of

 “Simultaneous High-Speed Video Laryngoscopy and Acoustic Aerodynamic Recordings During Vocal Onset of Variable Loudness: A Preliminary Study” 

Peak Woo

Manuscript ID: bioengineering-2904312

This manuscript  describes a new advancement of techniques for studying laryngeal behavior  and the biophysics of voice onset during speech production. The main advancement is combining high-speed video recording through a flexible fiberoptic laryngoscope with use of the Pentax  Phonatory Aerodynamic System (PAS), which is commonly used in the clinic to collect acoustic and aerodynamic information. Although this development is a useful advancement of techniques, the manuscript is not suitable for publication in its present form.  It needs a lot of editorial work to improve the presentation, including extensive proofreading to correct many obvious errors.  There is also at least one case in the Introduction  (lines 81-83) where the author falsely claims  a paper by himself and Baxter (reference 10) to be the first to perform high-speed filming through a fiberoptic laryngoscope in 2017.  However, references 5 and 13, from 1987 and 1988, respectively, already discussed high-speed filming through a fiberscope.

 A prime example of the need for extensive proofreading is Table 1. In order to fit the table onto the page in portrait mode, column widths are too small, so that even single-word headings are split into two rows (for example ‘subje’  and ‘ct’) when this could have been easily corrected by presenting it in landscape mode. The units for  ‘Pressure’ (column 8) should be (‘cm H2O’, not just ‘cm’} and for ‘Resistance’ should be ‘pressure/flowx1000’. In column 2 (‘sex’),  the entries are ‘male’ (lower-case m) and ‘Female’ (upper-case F) and the ‘Female’ is split into ‘Fema’ and ‘le’, even though much of the next column (‘age’) is blank and the age entries are far to the right of the column heading.  The same thing happens in  the next columns (‘toke’ and ‘n’ with entries ‘mod’ and ‘’al’) although there is a lot of blank space under ‘SPL’ and ‘dB’, which are not on top of the numeric entries in that column. In short, there has been gross lack of attention to detail in the construction of that table.

Another example of bad proofreading and attention to detail is in the name of the second of the three  levels of vocal intensity. It is ‘medium’ (line 24), ‘mobile’ (line 105), ‘middle’ (line 128), and ‘modal’ (line 131 and most other places). The word ‘modal’ was also used to describe pitch in the ‘Subjects instructions’ paragraph (lines 150-155), which ends describing the three voice levels as ‘soft, medium, and loud’.

The sentence  on lines 229-231 doesn’t seem to make much sense. The phrase ‘was noted’ appears twice on line 235.

On line 159, the acronym HSV (‘high-speed video’) is used without first defining it. In CT,fact it is used many times throughout the paper but is not defined until the Discussion section (line 359).

Figure 2 shows an example of a ‘PAS tracing’ that would not be of any use to a reader who isn’t already familiar with the PAS system.  It would be useful to identify the 4 waveforms, at least in the figure caption and to describe some details. What are the units on the left and what do the numbers in the white spaces above each trace represent?  The figure is said to represent the syllable ‘paa’ spoken at three loudness levels repeated twice, but it appears that the soft one was repeated twice the second time around.  If this is true, it should be mentionefsd in order to not confuse the reader.

Line 217, near the beginning of the Results section, should be given the sub-section heading 4.1 and line 259 should be 4.2 rather than 4.1. Line 310 should then be 4.3 rather than 4,2.

In Table 1, there are three males identified as subject 1 to 3 and three females identified  as subjects 4 to 6. Line 260 refers to ‘female subject 1’.  This is actually subject 5 and should be identified that way. The paragraph at the top of page 12 refers to female 3, who appears to be subject 6.  It also refers to Male 1.  Which male was that? Presumably, then, subject 4 is female 2.

Figures 11, 12,  and 13 show the same kind of data. Figure 11 is said to show data for subject 4, who is probably female 2 (see above). Lines 335-336  refer to “Figure 12 and Figure 13 for the two female subjects”, presumably meaning for the ‘other two female subjects’. One is identified on Figure 12 a Subject 3, who is a male, so it is probably means female subject 3, who is subject 6. The other is identified on Figure 13 as Subject 5, who was called female subject 1 (see above). Although all figures show the same kind of data, Figure 11 has no x-axis label and figures 12 and 13 have different x-axis labels.These differences should have been spotted when the manuscript was proofread.

 I also have more substantive criticisms or questions about the interpretation of the data, but I don’t have the energy to pursue these further at this point. Because I am not familiar with the details of the PAS system, I questions to what extent the flow measurements represent only the flow during phonation and are not affected by the flow during the prephonatory /p/ aspiration and I wonder whether the subglottal pressure, presumably measured during the /p/ closure, accurately represents the subglottal pressure during phonation.

Comments on the Quality of English Language

Moderate editing of English language required

Author Response

(The authors gave the same response as above.)

Reviewer 3 Report

Comments and Suggestions for Authors

This study uses flexible high-speed video laryngoscopy to analyze voice onset patterns, showing changes in vocal tract adjustments, glottal resistance, and harmonic energy with variable loudness in normal subjects.

My comments are as follows:

1.         It is generally believed that higher loudness usually corresponds to higher frequency, which can be clearly observed in certain subjects (especially #2, #6), but not necessarily in others, or not noticeable between soft and modal. Has the author provided any corresponding explanations?

2.         The change in resistance from modal to loud is much greater than from soft to modal. What prompts were given to the subjects regarding the intensity of phonation? What are the operational definitions?

3.         Are the data presented as averages or just single measurements? “The subject produced each token twice, followed by the next token.” Is there a selection of data? If presenting averages, the data should include content, and for multiple measurements, include mean±SD.

4.         Table 2 presents average data for females and males but does not make statistical comparisons, making it difficult for readers to discern meaningful differences.

5.         In paragraph 4.1, does female subject 1 refer to subject 4? Please clarify.

6.         Are non-laryngeal muscle movements during phonation observed through DKG? Is there a correlation with PAS? Factors such as resonance, training, and habits have a significant impact on non-laryngeal actions. Even with a limited number of patients in this study, are there any observations or explanations related to this?

Overall, the author provides a concrete description of the Simultaneous High-Speed Video Laryngoscopy and Acoustic Aerodynamic Recordings technique and offers insightful analysis on normal individuals, contributing to the future standardization of examinations and interpretations, making it highly readable and instructive.

Author Response

(The authors gave the same response as above.)

Round 2

Reviewer 1 Report

Comments and Suggestions for Authors

The manuscript has been improved considerably through revision. The introduction is more comprehensive and gives motivation for the study more clearly. The figures are now clearer. The procedure and results are more clearly presented. However, there are still some typos (perhaps they can be removed in publishing process), and some sentences are somewhat unclear or include unnecessary repetition. Below is a list that may help the author to do the final minor revisions.

Abstract

“… because due to viewing the larynx was viewed using rigid endoscopes”.

Please, revise. For instance, simply: because the larynx was viewed…

“We adapted a method to perform simultaneous high-speed video using nasal laryngoscopy with high-speed imaging while simultaneously acquiring…

Please, revise. For instance: to perform simultaneous high-speed naso-endoscopic video while simultaneously acquiring…

Introduction
3rd line from beginning, typo: “ues”- should be uses?

“The standard acquisition of phonation is to ask the subject to produce token obtained by asking the subject for phonation at the target fundamental frequency and sound amplitude”.

The sentence requires revision. For instance: The standard procedure to acquire phonation (to be studied) is to ask the subject to phonate at a target pitch and loudness.

“Fiberscope high speed imaging would be an ideal method for imaging of vocal fold vibration and such approaches have been published in prior studies.”

Here a reader expects to see references given.

“iDigital kymography tracings were obtained during natural speech; however, images

recorded using this technique require enhancement and manipulation. Despite the use of

image enhancement and noise reduction software, objective digital waveform analysis

cannot be performed due to high noise”.

“iDigital” is probably a typo and it should be digital.

Additionally, it is not clear what the paragraph refers to: The present study or some previous studies? Would it be clearer by using another tense: tracings have been obtained… (references)?

Tongue tension in mentioned in different parts of the manuscript. However, strictly speaking, how do you see tension? It is easier to understand that it is possible to see structures and changes in their position or in their form. Could you please somehow describe how you come to the conclusion that the tongue tension is increased.

Last sentence in the introduction: full stop needed.

Materials and methods

“one was an aspiring singer, the others were English speakers”. This brings a question: Was the aspiring singer a speaker of another language than English? Additionally, maybe it is more accurate to say “speaker of English” instead of “English speaker”.

Sound amplitude. I suggest to use sound pressure level (SPL), since that is what is measured.

Figure 2 caption: please add H2O (after “cm”).

Subject recordings

Page 5: “The three tokens were soft phonation at the modal pitch, the most comfortable sound am-

plitude at the medium loudness, and loud phonation at the modal pitch.”.

I suppose you asked the participants to produce different loudness levels, and did not measure SPL at the same time to choose the dB level, which the subject would regard as most comfortable. Thus my point is: Maybe it would be more accurate to say: the most comfortable loudness within the range of medium loudness levels.

2.2.2. Recording of aerodynamic data

“An adult flow head was used for all PAS flow recordings”.

What is a flow head? A flow mask? Or a flow transducer?

3. Analysis

3.1. Aerodynamic and Acoustic Analysis.

“For each token, the data were averaged between two tokens, Because there was also

variations within the token in the frequency, SPL, flow, and pressure, it was necessary to

take an averge for each token.”

Three typos in the sentence, and the formulation is not clear.

I recommend: There were variations within and between the tokens in the fundamental frequency, SPL, flow, and pressure. Therefore averages of the parameters were taken from the two repetitions of each token. – Is that what you mean?

“…the average sound amplitude (dB)…”

I suggest: the average sound pressure level, SPL (in dB)

3.2.

“A description of the video analysis for deriving the vibrogram waveform has been

published 1011.” Please, revise the reference notation.

3.3.

“The spectral peak values can be used to compare the spectral peak values for each vibrogram wave-

form for a specific token.”

I suggest: The spectral peak values from each vibrogram waveform can be used to compare different tokens.  Maybe you could tell already here what the FFT spectrum of a vibrogram waveform tells about vocal fold vibration. Please, add also references.

Page 7. “The necessary change in the resistance per decibel was more significant for modal-loud phonation than that for soft-modal phonation”.

More significant? It seems to refer to statistics. Would it be better to say: more remarkable or larger, instead of significant?

Table III: there is a p-value given only for the change from soft to modal, not for the change from modal to loud.

Moreover, the use of the word modal is confusing, since sometimes it refers to modal pitch and sometimes it is used as a synonym for medium (loudness). Please, unify the terminology used.

Page 8, 6th line from top:” it then assumed…” What assumed? Please, use ‘the glottis’ instead of ‘it’.

Figure 5: Sorry, I don’t understand why the down arrow here is ahead of right arrow? Do I misunderstand here something?

Figure 6. “closing time” – or is it closed time you mean?

11/17 end: “The need for excessive supraglottic adjustment before the onset of phonation was more

significant in males than females.”

I wonder how it is possible to say what is excessive. Also: Instead of significant I think that ‘remarkable’ or ‘noticeable would be better.

Figure 10 is a nice, clear picture. Could you please add in the figure caption what the arrows show.

Page 12:

“With louder phonation, a more significant energy peak was observed as a spectral peak at the first and second harmonics.”

I think that there is unnecessary repetition in the sentence. Could it be put like: In louder phonation the first and second harmonics of the FFT from vibrogram waveform were stronger. ?

 “This harmonic energy increase is consistent with the observation of short open and longer closed phases of vocal fold oscillation.”

Could you please add a reference to literature or is this an observation by the author while performing this study?

“With loud phonation, there is higher spectral energy at the first, second, and third harmonics of the fundamental frequency.. “

Please, revise, e.g. : In louder phonation, the first three harmonics gain in strength. (Please note that the ‘first harmonic’ is the fundamental itself.) There is also a typo: twice a full period in the end.

“This higher spectral energy peaks indicates a higher amplitude of vocal fold oscillations with loud phonation. “

The sentence requires revision. Moreover, what does the sentence mean? Does it mean that the spectral peaks are stronger or that there are higher harmonics visible  - or both? Could it be put:These stronger spectral peaks indicate…in loud phonation?

From voice source analysis it is known that the amplitude of the first harmonic corresponds to the vibratory amplitude of the vocal folds and the strength of the overtones corresponds to the glottal closing speed (Gauffin & Sundberg 1989).

“The higher energy peaks and energy at higher harmonics were consistent with the observation of higher acceleration of the vocal folds.”

Again I find this unclear. Moreover do you mean that you observe acceleration visually?

“More extraordinary spectral peaks with loud phonation were a consistent finding, as shown in Figure 12 and Figure 13 for the two female subjects. These figures show an increase in the F0 energy with an increase in sound amplitude. Increases in F1 and F2 energies with increasing sound amplitude were observed.”                      

I don’t understand what is meant by “More extraordinary spectral peaks”. Can you explain? Moreover, please tell in figure captions what the red line and black line mean. What you mean by “F1” and “F2”? Do you mean first and second harmonic? F1 and F2 mean first and second formant. If you refer to harmonics, please, use H1 and H2 or 2fo, 3fo etc.

Discussion, 5th line from beginning, typo: videdokymography

Page 14, instead of “normal” I recommend normophonic participants. Next paragraph:

“A steady-state delay occurs when there is prolonged vocal folds oscillate until a steady state is observed.” The sentence requires revision, e.g.  oscillation instead of “oscillate”.

Next paragraph: “vocal-fold” In other places there is ‘vocal fold’ so please unify the way of writing.

“This would have tabulated the time from glottis adduction to full steady state oscillation.” ? Should it be: This would have required tabulation of the time..?

“T wo, only six subjects sampled at three different loudness levels, the descriptive findings are the best that can be tabulated.” The sentence is incomplete and there is a typo: t wo

Should perhaps be: only six subjects were sampled (or recorded…?)…

Page 15

“The video correlated with the loud token is the prolonged delay from vocal fold closure to steady state.” Unclear sentence. Should it be: shows (instead of “is”)?

“Higher spectral peaks indicate higher acceleration and deceleration of the vocal folds, with an increase in amplitude and longer closing time of the vocal folds.”

Do you mean higher in frequency or amplitude? I mean: Can it be: Stronger spectral peaks… And increase in amplitude – do you mean louder phonation (increase in sound pressure level) or increased vibratory amplitude of the vocal folds? Please clarify.

Conclusion

“This is a preliminary study on a technique to obtain simultaneous HSV laryngoscopy

acquisition can be performed using acoustic and aerodynamic data sampling”.

The sentence requires revision. Maybe it could be put: This is a preliminary study on a technique

to allow simultaneous acquisition of nasoendoscopic HSV, acoustic and aerodynamic data.

“Future evaluations of vocal physiology should include visual, acoustic, and aerodynamic data to understand its natural function in both normal and pathological states”.

I recommend: Future evaluations should include…to understand vocal physiology in…

References: Need to be checked, there are various typos and other unclarities. For instance, the name Kiritani is twice written wrongly (references 5 and 12). Reference 15: typo:ef ficacy

Comments on the Quality of English Language

Some revision needed.

Author Response

(The authors gave the same response as above.)

Reviewer 2 Report

Comments and Suggestions for Authors

Improvements have been made in this revised version but there are still many problems. As before, many of them could have been dealt with by adequate proofreading.  For example, the author states th at he has gone through the manuscript and changed all uses of "modal" or "middle" to "medium" in referring to loudness levels but there are many "modals" left.  There are examples in figures 2, 9, 10, 11, 12, and 13 and their captions. There are 11 examples on page 7.  There is also a "middle" above Figure 2 on page 4 and a "modal" in the Figure-2 caption.

In several places, the units for pressure are given in cm.  It should be cm H2O.  That was used properly at the end of Section 3.1.

"Vocal fold" on the first line on page 6 should be "glottis".

An error that I missed in the original manuscript still remains. Section 4.2 begins with "Figures 4, 5, and 6 show ...". Those numbers should be 5, 6, and 7.

I don't understand the placement of the arrows in Figures 5, 6, and 7.  I assume that the time order of the images in these figures is left to right in a row and then top to bottom. The text for each of these figures states that the "right arrow  indicates just noticible onset of oscillation" and the "down arrow indicates beginning of steady state oscillation". In all 3 cases, the right arrow is at a later time than the down arrows.  Perhaps  the descriptions of the two type of arrows are reversed or perhaps the arrows are in the wrong places. "Noticible" should be  "Noticeable".  Should "onset of oscillation" be "onset of adduction"?

The term "token" is used to mean two different things. Sometimes it just means "an utterance". Other times, it means utterances at one of the three loudness levels. For example it is used both ways in the first sentence of Section 3.1.

There are many more similar minor errors that need to be corrected. The author needs to do a much better job of checking the text.

Comments on the Quality of English Language

The corrections of the English are not always appropriate. They should be gone over by someone who is familiar with the field.  For example, in the first sentence of the Abstract on page 1, "sound amplitude levels" should be "sound pressure levels". One page 2, line 2, stroboscopy should be described as a technique or a method, not an instrument. 

Author Response

(The authors gave the same response as above.)

Reviewer 3 Report

Comments and Suggestions for Authors

The author responded my comments properly and in great detail, and the article has improved significantly after being revised. I think the current version is very instructive and readable.

Author Response

Thank you for acknowledging the significance of our article. We will continue meaningful research based on your support.